# Interleukin-11: A Potential Biomarker and Molecular Therapeutic Target in Non-Small Cell Lung Cancer

**DOI:** 10.3390/cells11142257

**Published:** 2022-07-21

**Authors:** Jason Hongting Leung, Benjamin Ng, Wei-Wen Lim

**Affiliations:** 1Department of Cardiothoracic Surgery, National Heart Center Singapore, Singapore 169609, Singapore; 2National Heart Research Institute Singapore, National Heart Center Singapore, Singapore 169609, Singapore; benjamin.ng.w.m@nhcs.com.sg (B.N.); lim.wei.wen@nhcs.com.sg (W.-W.L.); 3Cardiovascular and Metabolic Disorders Program, Duke-National University of Singapore Medical School, Singapore 169609, Singapore

**Keywords:** interleukin-11, cytokines, non-small cell lung cancer, biomarkers

## Abstract

Non-small cell lung cancer (NSCLC) accounts for 85% of lung cancer and is a fast progressive disease when left untreated. Identification of potential biomarkers in NSCLC is an ongoing area of research that aims to detect, diagnose, and prognosticate patients early to optimize treatment. We review the role of interleukin-11 (IL11), a stromal-cell derived pleiotropic cytokine with profibrotic and cellular remodeling properties, as a potential biomarker in NSCLC. This review identifies the need for biomarkers in NSCLC, the potential sources of IL11, and summarizes the available information leveraging upon published literature, publicly available datasets, and online tools. We identify accumulating evidence suggesting IL11 to be a potential biomarker in NSCLC patients. Further in-depth studies into the pathophysiological effects of IL11 on stromal-tumor interaction in NSCLC are warranted and current available literature highlights the potential value of IL11 detection as a diagnostic and prognostic biomarker in NSCLC.

## 1. Brief Overview of NSCLC Management and Importance of Biomarkers in Clinical Context

Lung cancer is the second most commonly diagnosed cancer and is the leading cause of cancer death in the world [1]. The two main types of lung cancer are small cell and non-small cell lung cancer (NSCLC), characterized by the morphology of cancer cells under light microscopy. NSCLC accounts for 85% of lung cancer cases and has an overall 5-year survival of only 25%. Patients with early-stage NSCLC are anticipated to have a 10 month prognosis if left untreated [2], demonstrating its aggressive nature as an early progressing disease. Therefore, early recognition and diagnosis is of utmost importance to maximize patient survival.

The prognosis of NSCLC correlates largely with the extent of disease. In clinical practice, this is described by the tumor, node and metastasis descriptors and each combination corresponds to a prognostic American Joint Committee on Cancer stage grouping. Stage grouping forms the predominant means for patient stratification and the basis for evidence-based treatment modalities. Surgical resection is recommended for patients with operable stage I and stage II disease [3]. Adjuvant chemotherapy can improve patient outcomes for resected stage IB disease and above [4]. Stage IIIA disease usually involves multimodality management, whereas stage IIIB and above are generally managed non-surgically by chemoradiotherapy, targeted therapy and immunotherapy. Predictive biomarkers can help stratify lung cancer patients who better respond and benefit from specific targeted therapies and immunotherapies from those who do not. Testing for actionable mutations and immune biomarkers are now part of guideline-based management. Biomarkers for recurrence and resistance to treatment is an ongoing area of research and can potentially change surveillance and treatment practices.

Early detection of lung cancer has the potential to improve overall outcome by shifting the disease population to earlier stages corresponding to better prognosis. This goal is especially important in the context of NSCLC where patients are often first diagnosed at an advanced stage. Computed tomography (CT) is the current standard method for identification of lung nodules in lung cancer screening. The National Lung Screening Trial enrolled high-risk current and former smokers in the United States and demonstrated a mortality benefit of 20% for participants screened by CT compared to chest radiography. There was a substantial shift towards early identification of lower stage cancers at diagnosis and therefore eligibility for curative treatments in the CT screening arm [5]. The NELSON trial demonstrated a reduced cumulative rate ratio for death from lung cancer at 10 years of 0.76 (95% confidence interval 0.61–0.94) in the CT screening group compared to no screening [6]. Benefits of CT screening potentially extend to non-smoking populations [7]. This screening modality is not without limitations, which includes resource requisite, testing costs, radiation exposure risk with recurrent screening regimens and high rate of false positives resulting in overdiagnosis. When lung cancer is suspected radiologically, histological evaluation that relies on invasive percutaneous, bronchoscopic or surgical biopsy remains the necessary gold standard for definitive diagnosis. A portion of patients with no lung cancer unavoidably develop unnecessary complications as part of the diagnostic process. A recent systematic review found that CT guided percutaneous biopsy has a pneumothorax rate of 25.9%, and 6.9% required chest tube insertion [8]. Identification of suitable diagnostic biomarkers is therefore critical to supplement the goal of earlier detection and diagnosis with reduced risk, improved accuracy and efficiency.

Interleukin-11 (IL11) is a pleiotropic cytokine that has recently emerged as a tumor-promoting biomarker for cancer. In this narrative review, we discuss the mechanisms by which IL11 may promote NSCLC tumor growth and summarize the evidence regarding the diagnostic and prognostic utility of IL11 specifically in NSCLC. 

## 2. Interleukin-11: Member of the IL6 Family of Cytokines

Cytokines are small soluble secreted proteins that participate in autocrine, paracrine, and endocrine signaling to facilitate a diverse range of physiological functions including regulation of immunity, inflammation, cellular proliferation and cellular growth. They are often classified based on commonalities in structure, function or receptors. IL11 is classified in the IL6 family of cytokines that share commonality in the glycoprotein 130 (gp130) signaling receptor subunit, in combination with their cytokine-specific cognate receptors [9,10]. This family of cytokines and their cognate receptors have been regarded as important contributors to cancer biology (reviewed elsewhere in [11,12,13,14]), and may serve as potential biomarkers in disease progression. Similarly, other members of the IL6 family cytokines are known to be implicated in NSCLC as well (Table 1). 

## 3. Interleukin 11 Drives Pulmonary Fibrosis and Inflammation

IL11 was initially implicated in several inflammatory lung diseases such as asthma and tuberculosis infection. However, recent evidence has demonstrated that IL11 is an important determinant of fibrosis and chronic inflammation in the lung (reviewed in [28]). Pulmonary epithelial cells and fibroblasts express high levels of IL11RA and are prominent cellular sources of IL11 in response to respiratory infections. Hence, it is proposed that IL11 acts in an autocrine/paracrine fashion in response to pathogens or after tissue injury. IL11 is upregulated in pulmonary fibroblasts from patients with idiopathic pulmonary fibrosis (IPF), a form of progressive fibrosing interstitial pneumonia characterized by an excess of activated myofibroblasts, and its elevated expression in the IPF lung is associated with fibrosis and disease severity [29]. In vitro studies on non-transformed pulmonary cells showed that IL11 triggers the proliferation and transformation of quiescent fibroblasts into collagen-producing invasive myofibroblasts [29], and induces an EMT program in epithelial cells. Pharmacological inhibition of IL11 or fibroblast-specific blockade of IL11 signaling reduced fibroblast invasion in vitro and reversed pulmonary fibrosis and inflammation in a murine model of IPF [29,30]. Interestingly, the prevalence of lung cancers in patients with IPF is significantly higher than the general population, with NSCLC being the predominant type of lung cancer in these patients [31]. Furthermore, IPF patients with lung cancer have increased mortality rates as compared to IPF patients without a lung cancer diagnosis [32]. However, it is as yet unclear whether IL11 plays an active role in lung cancers in IPF.

## 4. Interleukin 11 Is a Tumor-Promoting Cytokine in NSCLC

IL11 was first isolated in 1990 from a primate bone marrow-derived stromal cell line and was identified to possess hematopoietic and thrombopoietic properties [33]. Subsequently, IL11 was evaluated as a potential therapy for chemotherapy-induced thrombocytopenia among cancer patients [34,35]. The need to determine suitability of IL11 treatment for thrombocytopenia in cancer patients motivated early studies evaluating its effects on cancer cells.

Early studies involving lung cancer cells concluded that IL11 did not promote tumor cell growth. Soda et al. harvested tumor cells from a heterogenous group of clinical specimens and subjected the cells to recombinant human IL11 (rhIL11) [36]. This study defined tumor growth stimulation as a >150% survival increase in tumor colony-forming units and found that 97% of the specimens across a variety of cancers tested were not stimulated by rhIL11. Saitoh et al. utilized a murine model of lung cancer and showed that rhIL11 inhibited proliferation in vitro while not affecting the anti-tumor effects of carboplatin, mitomycin and etoposide in mice [37]. Treating Calu-1 cells, a NSCLC epithelial cell line, with IL11 seemed to reduce DNA synthesis but not significantly [38]. Taken together, these early studies suggest that IL11 was more likely an inhibitor, rather than a stimulator, of tumor growth. 

Later studies, however, identified IL11 as a tumor-promoting cytokine instead [39,40,41,42]. It is now known that rhIL11 does not activate mouse IL11 receptor (IL11RA), but competitively inhibits binding of endogenous murine IL11 instead [43,44]. Hence, the use of rhIL11 on murine lung cancer is not expected to recapitulate the effect of human IL11 on human lung cancer. It is likely due to the differential effects of cross- and same-species recombinant IL11 that a later study utilizing same-species rhIL11 concluded IL11 to be tumor promoting [45], while an earlier study that used cross-species rhIL11 concluded otherwise [37]. Zhao et al. provided direct evidence that IL11 promotes tumor growth using lentivirus-mediated IL11 overexpression and knockout in A549 and H1299 lung cancer cell lines [46]. In subsequent studies, the same group demonstrated that hypoxia downregulated miR-495 and miR-5688 to enhance IL11 expression and promote tumor progression [47], showing that IL11 is especially important for tumor growth during tumor hypoxic conditions. In support of the finding that IL11 promotes cancer pathology, multiple clinical studies have now associated IL11 to poorer prognosis in lung cancer (reviewed below).

The sources of IL11 in lung tumors comprise tumor cells, as well as stromal cell types such as fibroblasts [48,49], airway smooth muscle [50] and epithelial cells [51]. Fibroblasts are important components of the tumor microenvironment and are known to modulate tumor behavior. Clinical studies in NSCLC patients show that fibroblast activation correlates with poorer survival [52,53,54]. In human lung tumor tissue samples, IL11 is found to co-localize with ACTA2-positive cells (a marker of myofibroblasts) [55], and single-cell RNAseq analyses showed that *IL11* gene was expressed in both tumor cells and fibroblasts [56]. IL11 secretion by fibroblasts is an important medium for NSCLC tumor growth and mediates fibroblast-tumor cell crosstalk. Microvesicles from A549 and HTB177 cell lines upregulate IL11 expression from fibroblasts to synergistically promote disease progression [57]. In addition to promoting tumorigenesis, secretion of IL11 from fibroblasts is one mechanism by which NSCLC tumors resist cisplatin treatment [55]. The findings that cancer-associated fibroblasts secrete IL11 to promote tumor growth and confer chemoresistance has also been observed in other cancers including gastric cancer [41,42]. 

IL11 signaling can occur either via classical signaling or trans-signaling. During classical signaling, IL11 binds to its specific receptor IL11RA, which leads to dimerization of the signal transducing gp130 subunit and activation of downstream signaling. IL11RA is differentially expressed in NSCLC and has been targeted in a preclinical study to reduce tumor growth [58]. In trans-signaling, soluble IL11RA are generated from mRNA splice variants or from proteolytic cleavage of the membrane-bound receptor complexing with the cytokine extracellularly [59]. This complex then binds to any gp130, which is ubiquitously expressed on cell surfaces, without requiring the cytokine-specific receptor potentially expanding the repertoire of effector cells. IL6 trans-signaling has been reported to be important in KRAs-driven NSCLC [60], but it is yet uncertain whether IL11 functions similarly. Downstream of cytokine-receptor binding, JAK/STAT [39,40,61], PI3K/AKT, and Ras/ERK pathways can be activated resulting in cell proliferation, inhibition of pro-apoptotic proteins, activation of anti-apoptosis proteins, and angiogenesis [62,63]. Activation of these pathways has been proposed as a prognostic outcome predictor for NSCLC [64,65,66,67]. 

IL11 signaling in NSCLC has been investigated mostly in the A549 cell line as a model for lung adenocarcinoma (LUAD), the most common subtype of NSCLC. IL11 has been consistently reported to activate STAT3 [45,46,55], and upregulates anti-apoptotic pathways driven by BCL2 and Survivin [55]. Recently, IL11 stimulation has also been found to activate ERK and p90RSK to inhibit LKB1/AMPK and increase mTOR [68]. LKB1 is an important tumor suppressor for which inactivating mutations are prognostic for poorer outcome and predictive for treatment failure in NSCLC (reviewed in [69,70]). Activation of the mTOR pathway is commonly seen in lung cancer and drives pathology [71]. Potentially, IL11 stimulation can result in suppression of LKB1/AMPK in tumors that express functional LKB1 protein to lead to worse outcomes.

IL11 can also drive cancer growth by promoting epithelial-mesenchymal transition (EMT) [45,46,72]. EMT describes the transition of epithelial cells possessing apical-basal polarity and cell-cell adhesion properties into non-polarized mesenchymal cells with increased migratory and invasive properties. This process allows epithelial tumors to invade into the extracellular matrix-rich stroma, and is an important process for tumor metastasis. Increased EMT has been associated with poorer prognosis in NSCLC patients [73]. 

## 5. Sources of Biomarkers in NSCLC

Biomarkers are measurable characteristics of normal and pathogenic processes or exposure to interventions [74]. Sources for molecular biomarkers in NSCLC may include peripheral blood, bronchoalveolar lavage, breath exudate, lung tissue, pleural fluid, urine, sputum and saliva. These sources differ in feasibility, reproducibility, and procedural risk profile. Peripheral blood is readily accessible and is useful for comparisons between healthy, disease and treated populations and is also suitable in longitudinal studies measuring response to therapy. However, the sensitivity, specificity and reproducibility of blood biomarkers is limited by production by sources other than tumor, volume of distribution, and biomarker metabolism and turnover. Alternatively, molecular biomarkers identified via lung tumor tissue most directly reflect true tumor biology. However, procurement of tissue samples requires invasive methods such as diagnostic biopsy samples, post-mortem collection, or from surgical resection. Specimens from surgical resection provide the greatest quantity of tumor tissue and allows adjacent histologically normal tissue to be collected concurrently for comparisons. 

## 6. IL11 Differential Expression in NSCLC Tumor Tissue

The Cancer Genome Atlas (TCGA) database is a commonly referenced publicly accessible dataset containing RNAseq data for tumor samples from NSCLC patients, including a proportion of matched adjacent normal samples [75]. The Genotype-Tissue Expression (GTEx) project is a publicly available resource that provides genotype and expression data for tissue from different sources, including those from non-cancerous lung [76]. Based on the UCSC Xena platform [77,78], which utilizes these datasets collectively, *IL11* mRNA expression is elevated in tumor tissues as compared to adjacent normal or tissues from donors without lung cancer (Figure 1).

Similar results of elevated *IL11* mRNA expression by RT-qPCR in tumor tissues have also been noted in several small studies. Zhao et al. found increased *IL11* mRNA expression in a small cohort comprising 18 NSCLC against five normal lung tissue samples [46]. Subsequently, they demonstrated elevated *IL11* mRNA expression in mixed stage (stage I to IV) lung tumor tissue compared to adjacent normal tissue in a cohort of 28 patients, with the majority of patients having a tumor size ≥5 cm (71%) and nodal metastases (79%) [47]. Wang et al. found higher *IL11* mRNA in LUAD tumor tissue compared to adjacent normal tissue in a small cohort of 10 patients [79]. Brooks et al. observed a non-statistical increase in *IL11* gene expression in a cohort of 24 LUAD patients when compared to five LUAD-free individuals [60]. 

Compared to mRNA expression, protein expression more directly relates to cytokine biological function. Unfortunately, IL11 protein expression in lung cancer tissue is rarely reported. Multiple studies have relied on mass spectrometry to identify protein biomarkers in NSCLC tumor tissue but IL11 was not identified as a differentially expressed protein [80,81,82,83,84,85,86,87]. These negative findings from quantitative proteomics studies contrast with the findings of *IL11* mRNA differential expression that is frequently reported. One possibility is that IL11 protein expression may be discordant with mRNA expression. Chen et al. reported that only 22% of proteins had significant positive correlation to their mRNA in lung adenocarcinoma tissue [82] and discordance between IL11 mRNA and protein is biologically possible through post-transcriptional regulation [47,88,89]. An alternative explanation is that the methods used for mass spectrometry are unable to detect IL11, a secretory cytokine, in low abundance in tissue. This alternative explanation is supported by a study that demonstrated increased IL11 protein in lung cancer tissue as quantified by ELISA [55]. 

## 7. Detection of IL11 for Diagnosis of NSCLC

To the best of our knowledge, there are two studies that describe the utility of IL11 as a diagnostic biomarker (Table 2). Pastor et al. recruited a prospective cohort of 369 patients for which use of diagnostic biomarkers in bronchoalveolar lavage fluid (BALF) can potentially facilitate early detection of lung cancer [90]. Among 80 cytokines and growth factors, inflammation-related protein array analyses identified IL11 expression to be differentially increased in LUAD BALF samples in an initial discovery cohort, and subsequently validated in two separate exploratory and diagnostic cohorts. This study demonstrated that BALF IL11 expression was largely restricted to patients with LUAD, with or without chronic obstructive pulmonary disease (COPD). Despite the eventual diagnosis at stage III or IV in the majority of the LUAD cohort, the diagnostic performance of BALF IL11 remains similar between the subgroups even for the early stages. Interestingly, BALF IL11 expression was not increased in squamous cell carcinoma (SCC), a type of NSCLC of epithelial origin, which may suggest cell type-selective IL11 expression in NSCLC. In another study, Wu et al. measured IL11 protein concentration in serum and exhaled breath condensate in NSCLC cases compared to healthy donors and found increased IL11 expression in NSCLC patients even at early-stage disease [91]. Additionally, serum IL11 concentrations were incrementally elevated with progressive disease stages which correlated with tumor size, stage, presence of metastases, and degree of differentiation. Taken together, these studies suggest the potential of IL11 as a diagnostic biomarker in NSCLC patients; however, its contribution to clinical management is currently unclear.

Blood is one of the more readily accessible sources for biomarker discovery in the clinic, but several potential challenges exist with utilizing circulating IL11 concentrations as a diagnostic biomarker for lung cancer. Cytokines and interleukins are often present in the lowest concentrations in circulation [92,93], which may be difficult to detect specifically and sensitively. For example, IL11 was detected in half of the patients with a primary lung disorder (tuberculosis, lung cancers and pneumonia) in pleural effusions but not in peripheral blood [94]. Furthermore, in patients with a broad spectrum of cancers, those with detectable serum IL11 had worse survivability compared to those without [95]. Even in late stage III and IV NSCLC patients, Agulló-Ortuño et al. found considerably low plasma IL11 levels of 12.24 pg/mL and 10.60 pg/mL in LUAD and SCC respectively [96], which may require more sensitive assay methods for accurate assessment. Lastly, circulating IL11 concentrations may also be affected by other pathologies unrelated to lung cancer (Table 3). Taken together, several considerations must be made when utilizing blood-based IL11 levels as a biomarker in NSCLC: (1) Not all lung cancer patients have detectable circulating IL11 levels possibly reflecting disease severity, (2) site of assessment appears to be most accurate at the disease source, and (3) concomitant diseases can complicate interpretation of blood IL11 concentrations. 

## 8. Quantification of IL11 Expression for Prognostication of NSCLC

Accurate prognostication is important for identifying patients who may benefit from adjuvant or neoadjuvant therapy. Using molecular biomarkers in addition to clinical data can potentially allow patients to be stratified into risk groups with greater accuracy [106,107]. Numerous studies have identified *IL11* mRNA in NSCLC lung tissue taken at the time of surgical resection to be prognostic of overall survival, both by itself and among other genes as part of a prognostic signature (summarized in Table 4).

In addition to published literature, there are multiple online tools drawing on publicly accessible datasets to facilitate comparisons of clinical and molecular data [112,113,114,115]. A statistically significant difference in patient survival was observed between high and low tissue *IL11* mRNA expression among LUAD and SCC patients in the TCGA Pan-Cancer Atlas dataset and high *IL11* expression was associated with poorer survival (Figure 2). 

## 9. Conclusions

Earlier studies investigated the effects of IL11 in cancer as a potential treatment for chemotherapy-induced thrombocytopenia among cancer patients. These studies, which utilized cancer cell lines and non-species-matched recombinant IL11 in mice, initially suggested that IL11 was unlikely to promote cancer and perhaps inhibited tumor progression. Recently, there is accumulating evidence suggesting that IL11 is an important tumor-promoting cytokine that that has both diagnostic and prognostic value in patients with NSCLC. Multiple in vitro studies confirm that IL11 activates known tumor-promoting signaling pathways and clinical studies link increased IL11 expression to poorer prognosis. Measurement of IL11 RNA or protein in blood, BAL and tissue may aid diagnosis and prognostication in patients with NSCLC, although feasibility and utility should be considered. Further research into the molecular and physiological effects of IL11 in NSCLC can reveal novel therapeutic targets.

## Figures and Tables

**Figure 1 cells-11-02257-f001:**
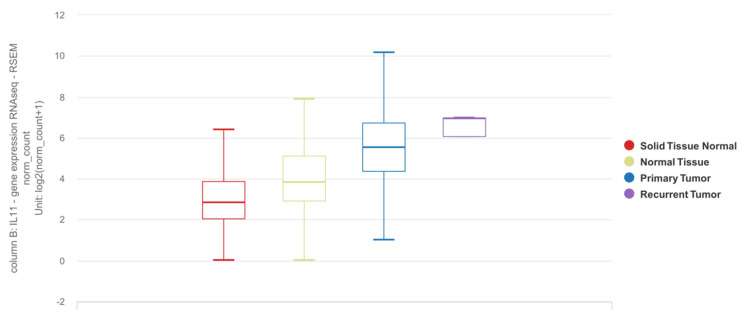
Differential expression of *IL11* mRNA in lung adenocarcinoma (LUAD) and squamous cell carcinoma (SCC) based on the TCGA TARGET GTEx dataset. Solid tissue normal: adjacent normal tissue from TCGA dataset (n = 109). Normal tissue: tissue from subjects with no lung adenocarcinoma from GTEx dataset (n = 288). Primary tumor: lung tumor tissue (LUAD or SCC) from TCGA dataset (n = 1011). Recurrent tumor: lung tumor tissue from recurrence (n = 2). The results shown here are in whole or part based upon data generated by the TCGA Research Network [75] and GTEx [76] databases using the UCSC Xena online platform [77,78]. https://xenabrowser.net was accessed on 26 April 2022.

**Figure 2 cells-11-02257-f002:**
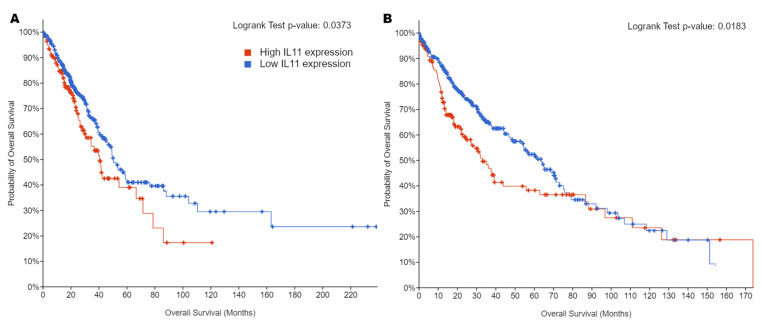
Kaplan–Meier curves for overall survival in patients with either high or low tumor IL11 mRNA expression in the TCGA PanCancer Atlas datasets for (**A**) LUAD (n = 501) and (**B**) SCC (n = 478). A cutoff of z > 0 was used for high expression and z ≤ 0 as low expression. Logrank tests show statistically significant differences between the high and low expression groups. The results shown here are based upon data generated by the TCGA Research Network [75,116], generated with cbioportal (https://www.cbioportal.org) [113,117] that was last accessed on 26 April 2022.

**Table 1 cells-11-02257-t001:** Other IL6 family cytokines and components of the specific receptors associated with NSCLC.

Cytokine	Receptors	References
IL6	IL6R, gp130/IL6ST	[15,16,17,18,19,20,21,22]
IL-31	IL31Rα, OSMR	[23]
LIF	LIFR/LIFRα, gp130/IL6ST	[24]
OSM	OSMR/OSMRβ, gp130/IL6ST, LIFR	[25,26]
CLCF1	CNTFR, LIFR, gp130/IL6ST	[22,27]

**Table 2 cells-11-02257-t002:** Studies identifying IL11 as a diagnostic biomarker.

Study	Recruited Population	Comparison	Sample Type	Diagnostic Biomarker	Assay	Receiver Operator Curve and Test Metrics					
						AUC (95% CI)	Cutoff (pg/mL)	Sensitivity (95% CI)	Specificity (95% CI)	PPV (95% CI)	NPV (95% CI)
Pastor et al. [90]	Age > 40 yrs, current or ex-smokers of 30 pack-years, evaluated for hemoptysis or pulmonary nodule or mass, excluding those with prior diagnosis of malignancy, active tuberculosis, history of drug abuse or other inflammatory disease apart from COPD	LUAD vs. non-LUAD—First validation cohort (n = 149)	BALF	IL11 protein	ELISA	0.93 (0.90–0.97)	42.0	90.2 (79–95.7)	88.7 (90.6–93.5)	80.7 (68.7–88.9)	94.5 (87.8–97.6)
		LUAD vs. non-LUAD—Second validation cohort (n = 160)	BALF	IL11 protein	ELISA	0.95 (0.92–0.98)	42.0	90.6 (79.7–95.9)	83.0 (86.8–87.7)	60.8 (49.7–70.8)	96.8 (92.7–98.6)
Wu et al. [91]	NSCLC patients with no history of radiochemotherapy, immune-targeted therapy or surgery (n = 91 for serum, of which 63 have LUAD and 28 have SCC; 64 for EBC)	Healthy volunteers without acute or chronic infectious diseases, vital organ diseases, or genetic family tumor history (n = 72 for serum; 63 for EBC)	Serum	IL11 protein	ELISA	0.93 (0.88–0.97)	126.1	75.0	100.0	NR	NR
			EBC	IL11 protein	ELISA	0.78 (0.69–0.86)	21.5	78.1	79.4	NR	NR

BALF: bronchoalveolar lavage fluid; EBC: exhaled breath condensate; LUAD: lung adenocarcinoma; SCC: Squamous cell carcinoma; COPD: chronic obstructive pulmonary disease; AUC: Area under curve; PPV: positive predictive value; NPV: negative predictive value; NSCLC: non-small cell lung cancer; NR: not reported.

**Table 3 cells-11-02257-t003:** Pathologies where IL11 from peripheral blood has been reported to be increased in human subjects.

Pathology	Comparison	Source	Findings	Reference
Polycythemia vera	Healthy	Plasma	Increased	[97]
Rheumatoid arthritis with or without interstitial lung disease	Healthy	Serum	Increased in rheumatoid arthritis, more so with concomitant interstitial lung disease	[98]
Congestive heart failure	Healthy	Plasma	Increased	[99]
Severe pancreatitis	Mild pancreatitis	Serum	Increased	[100]
Breast cancer metastatic to bone	Primary breast cancer and healthy controls	Serum	Increased compared to healthy controlsCorrelated with shorter disease-free survival	[101]
Pancreatic cancer	Healthy	Plasma	Increased Correlated with survival.	[102]
Gastric cancer	Chronic superficial gastritis and chronic atrophic Gastritis	Serum	Increased in gastric cancer > chronic atrophic gastritis > chronic superficial gastritis	[103]
Preeclampsia	Normal pregnant gestation-matched control	Serum	Increased	[104]
Thoracic aortic dissection	Non-aortic dissection patients presenting with chest pain	Plasma	Increased	[105]

**Table 4 cells-11-02257-t004:** Studies identifying *IL11* as a prognostic biomarker.

Study	Year	Training Cohort	Validation Cohort (s)	Cancer Type	Prognostic Signature	Findings
Kratz et al. [107]	2012	Non-squamous NSCLC (n = 361)	Stage I non-squamous NSCLC (n = 433), and stage I-III non-squamous NSCLC (n = 1006)	Non-squamous NSCLC	11 Target genes (*BAG1*, *BRCA1*, *CDC6*, *CDK2AP1*, *ERBB3*, *FUT3*, *IL11*, *LCK*) and 3 reference genes (*ESD*, *TBP*, *YAP1*)	(1) Risk as identified by the 14 gene-expression assay was a statistically significant predictor of overall survival.
Watza et al. [108]	2018	NSCLC patients without history of bronchiectasis or cystic fibrosis (n = 280)	TCGA Lung SCC and TCGA LUAD datasets (n = 1026)	NSCLC	23 genes involved in the interleukin signaling pathway, including *IL11*	(1) Interleukin signaling pathway was one of three pathways that was significantly associated with survival out of 48 immune-centric pathways evaluated. (2) 23 genes were identified as drivers of the interleukin pathway enrichment, which included *IL11*. (3) Higher expression of *IL11* had worse overall survival in both cohorts
Wang et al. [79]	2020	TCGA LUAD dataset (497 LUAD tissues, 54 normal lung tissues)	n/a	LUAD	6 genes (*CRABP1*, *IGKV4-1*, *IL11*, *INHA*, *LGR4*, *VIPR1*)	(1) *IL11* expression is prognostic of survival. (2) Cytokine-cytokine receptor pathways, JAK-STAT signaling pathways were among the top 5 most significantly enriched pathways by differentially expressed immune-related genes. (3) *IL11* is differentially expressed in tumor tissue compared to adjacent normal tissue.
Fan et al. [109]	2021	TGCA LUAD dataset (n = 464)(majority stage I and II)	GSE13213 (n = 117), GSE30219 (n = 85), GSE31210 (n = 226), GSE72094 (n = 420)(majority stage I and II)	LUAD	5 genes (*IL7R*, *IL5RA*, *IL20RB*, *IL11*, *IL22RA1*)	(1) *IL11* is differentially expressed (by mRNA) in tumor versus normal tissue, and is prognostic of survival when used in a 5 gene signature model. (2) These 5 genes were selected as they were differentially expressed and prognostic and thought to play the most important role in LUAD
Chen et al. [110]	2021	TCGA LUAD (535 LUAD tissues, 59 normal lung tissues)GSE161116 (9 LUAD tissues, 9 LUAD brain metastasis tissues)	n/a	LUAD	6 genes (*TNFRSF11A*, *MS4A2*, *IL11*, *CAMP*, *MS4A1*, *F2RL1*)	(1) *IL11* is differentially expressed in tumor tissue compared to normal tissue in the TCGA database and is an independent factor affecting prognosis. (2) *IL11* expression has diagnostic value for brain metastasis
Peng et al. [111]	2021	GSE161116 (13 lung tumor tissues, 15 brain tissues), GSE747706 (18 lung tumor tissues, 18 normal tissues), GSE21933 (21 lung tumor tissues, 21 normal tissues) datasets	n/a	NSCLC andBrain tumor	n/a	(1) 20 genes (including *IL6* and *IL11*) are differentially expressed in both brain metastasis (compared to NSCLC lung tumor) and lung tumor (compared to normal lung). (2) High *IL11* expression is linked to poorer overall survival.

## Data Availability

The authors declare that all data supporting the findings of this study are included in the article or generated from publicly available datasets and online tools. Figure 1 was generated using the publicly available UCSC Xena online exploration tool and the TCGA TARGET GTEx dataset available from http://xena.ucsc.edu (last accessed on 26 April 2022). Figure 2 was generated using the publicly available cBioPortal online tool and the Lung Adenocarcinoma (TCGA, PanCancer Atlas) and Lung Squamous Cell Carcinoma (TCGA, PanCancer Atlas) datasets available from https://www.cbioportal.org (last accessed on 26 April 2022).

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
