# Peer review of "Interleukin-11: A Potential Biomarker and Molecular Therapeutic Target in Non-Small Cell Lung Cancer"

_cells, 2022, doi:10.3390/cells11142257_

Round 1
Reviewer 1 Report
Leung et al ‘Interleukin-11: a potential biomarker and molecular therapeutic target in non-small cell lung cancer’
An interesting review of IL11 in NSCLC as well as a more broad overview of IL11 in other cancers.
Overall this manuscript is very well written and easy to understand, although there is a tendency to have very long sentences. The manuscript would be improved by editing to remove sentences that run into a 4th line.
Minor comments:
Line 18-19 – don’t think you need this sentence (‘several reviews… context of NSCLC’) in the abstract. In the intro is fine.
Line 43-47 – consider making this into multiple sentences, it is a bit long at the moment.
Line 47-50 – this sentence is also a bit confusing -consider 2 sentences and expanding on eg. how predictive biomarkers assist decision making. Do the biomarkers indicate prognosis – ie can identify early vs late disease?
Line 56 – what country?
Line 113 – stromal cells from what tissue?
Line 276 – should the word ‘only’ be removed? Found in half of pleural effusions but not detected in peripheral blood?
Reviewer 2 Report
Dear Authors,
the manuscript sounds interesting and good. Please check English language to revise, and please add more to Conclusion.
